# A Systematic Review of the Current Measures of Theory of Mind in Adults with Schizophrenia

**DOI:** 10.3390/ijerph18137172

**Published:** 2021-07-04

**Authors:** Ya-Chin Yeh, Chung-Ying Lin, Ping-Chia Li, Chi-Fa Hung, Chun-Hua Cheng, Ming-Hui Kuo, Kuan-Lin Chen

**Affiliations:** 1Institute of Allied Health Sciences, College of Medicine, National Cheng Kung University, Tainan 701401, Taiwan; yehyachin@ms.szmc.edu.tw (Y.-C.Y.); cylin36933@gmail.com (C.-Y.L.); 2Department of Occupational Therapy, Shu-Zen Junior College of Medicine and Management, Kaohsiung 821004, Taiwan; 3Department of Occupational Therapy, College of Medicine, National Cheng Kung University, Tainan 701401, Taiwan; 4Department of Public Health, National Cheng Kung University Hospital, College of Medicine, National Cheng Kung University, Tainan 701401, Taiwan; 5Department of Occupational Therapy, I-Shou University, Kaohsiung 824005, Taiwan; pingchia@isu.edu.tw; 6Department of Psychiatry, Kaohsiung Chang Gung Memorial Hospital and Chang Gung University College of Medicine, Kaohsiung 833401, Taiwan; chifa@cgmh.org.tw; 7College of Humanities and Social Sciences, National Pingtung University of Science and Technology, Pingtung 912301, Taiwan; 8Department of Occupational Therapy, Kaohsiung Municipal Kai-Syuan Psychiatric Hospital, Kaohsiung 802211, Taiwan; kiwin8070@gmail.com (C.-H.C.); ming14214@yahoo.com (M.-H.K.); 9Department of Physical Medicine and Rehabilitation, National Cheng Kung University Hospital, College of Medicine, National Cheng Kung University, Tainan 701401, Taiwan

**Keywords:** COSMIN, measure, mentalization, schizophrenia, theory of mind

## Abstract

Adults with schizophrenia usually have impairments in theory of mind (ToM), which subsequently cause them problems in social interaction. Therefore, it is important for healthcare providers to assess their ToM using adequate measures. This systematic review evaluated current ToM measures (or ToM tasks) for adults with schizophrenia and summarized their specific characteristics, including the concept and construct, administration, and psychometric properties. From a review of 117 articles, 13 types of ToM tasks were identified, and the findings from these articles were qualitatively synthesized. The results showed that ToM tasks are diverse in their presentation modalities, answer modes, strategies of controlling cognitive confounders, and scoring. Most tasks employ cognitive and affective dimensions and target a specific, single ToM concept. The present systematic review found that psychometric evidence supporting the ToM tasks, such as internal consistency, test–retest reliability, unidimensionality, and convergent, criterion, and ecological validities, is insufficient. Based on the results, we propose several principles for selecting appropriate ToM tasks in practice, e.g., selecting a task with multiple ToM concepts, or an exclusive ToM construct containing the cognitive and affective dimensions. Moreover, future studies are needed to provide more psychometric evidence on each type of ToM task applied in people with schizophrenia.

## 1. Introduction

Theory of mind (ToM) refers to the ability to perceive and reason about other people’s opinions, beliefs, intentions, or feelings [1]. With ToM, an individual can predict others’ behaviors and then make appropriate responses in diverse social contexts [2]. Impairments of ToM have been reported in several mental illnesses, such as schizophrenia, autism spectrum disorders, major depression, bipolar disorder, borderline personality disorder, and Parkinson’s disease [3,4,5,6,7]. However, the ToM performance in people with schizophrenia reveals specific features that are distinct from those in other illnesses. ToM impairment has been deemed a state-mediated trait marker in schizophrenia across different illness states [8,9,10,11,12] and evidenced by the connection between the genetic risk variants and key neural-network mediating ToM [13,14]. ToM impairment in adults with schizophrenia causes them social difficulties even more than their neurocognitive function or symptom severity does [15,16,17,18]. Fortunately, it has been evidenced that the ToM impairment in people with schizophrenia can be improved by mentalizing-based interventions [19,20,21]. Measuring ToM in people with schizophrenia is crucial because ToM has been identified as an important target and outcome measure for psychosocial treatment [22,23,24].

ToM constructs can be divided into the cognitive and affective dimensions. Cognitive ToM pertains to inferences about thoughts, which is linked with social knowledge and reasoning abilities. Affective ToM pertains to inferences about feelings, which is dependent on the ability to apply one’s own emotional experience. Both dimensions of ToM are linked to distinct ToM processes with dissociated neuronetworks [25]. Abu-Akel and Shamay-Tsoory [25] have concluded that cognitive ToM is mediated by the dorsal stream of the ToM neuronetworks, such as the dorsomedial prefrontal cortex, dorsal anterior cingulate cortex and dorsal striatum; affective ToM, by the ventral stream, such as the ventromedial and orbitofrontal cortices, ventral anterior cingulate cortex, amygdala and ventral striatum. Thus, they present distinguishable behaviors in performing tasks and cause differentiated patterns of ToM performances between the healthy and clinical populations. Specifically, healthy individuals and people with schizophrenia reveal differentiated patterns of ToM performances on cognitive and affective ToM. Healthy individuals tend to make more errors on cognitive than on affective inferences, while people with schizophrenia have demonstrated a less prominent trend [26].

Apart from cognitive and affective ToM, ToM concepts have been proposed to understand the many facets of ToM [27,28,29]. ToM concepts are varied in their content and complexity. The basic ToM concepts involve understanding the other’s mental representations, e.g., understanding that an individual’s belief or representation about the world may contrast with reality (first order false-belief), or that an individual may lie to get what he/she wants (deception). The more advanced ToM concepts involve complex recursion, e.g., understanding someone’s false-inference about another’s thinking or feeling (second order false-belief); metapragmatic skills, e.g., an individual telling white lies to avoid hurting the feelings of the listener (white lies); pragmatic skills, e.g., understanding the other’s ironic remarks (irony). The ability to understand two mental states such as the speaker’s false-belief and the listener’s negative feelings in a faux pas situation is also included [30]. With the identified ToM concepts, the levels of ToM impairments or development of ToM can be differentiated.

Taken together, the perspectives of cognitive–affective ToM and ToM concepts can be adopted to understand various measures of ToM for people with schizophrenia. Regarding the perspective of cognitive–affective ToM, some measures assess only one dimension of ToM. For example, the False Belief picture sequencing task and Eye’s test only assess cognitive and affective ToM, respectively [31,32]. Many ToM measures with both cognitive and affective dimensions have been developed, such as the Faux Pas task [30]. As regards ToM concepts, some assessments include only one concept, such as ironic or hinting remarks [33,34], and some contain multiple concepts, such as the Strange Stories, which involve the concepts of double bluff, mistakes, persuasion, and white lies [35].

In addition, the presentation modalities, answer modes, control questions or items, and scoring are also different across ToM measures. For example, the ToM tasks might be presented in the modalities of verbal stories, visual pictures, or movies; employ open-ended or multiple-choice questions; employ control questions or items to ensure correct comprehension and memory of the task content; or be scored with dichotomous or polytomous ratings. These differences among the ToM tasks may confuse practitioners and researchers, especially those who are just entering this specific realm and want to examine ToM in people with schizophrenia. Therefore, synthesized information on these characteristics of the extant ToM tasks may assist potential ToM task users in better understanding these tasks and subsequently selecting appropriate ToM tasks for their use.

Indeed, the importance of assessing ToM in people with schizophrenia has been well documented, and different measures have been established in this field [28]. To inform mental health practitioners about the latest evidence on the psychometric properties of the available ToM tasks, this systematic review aimed to evaluate the current ToM measures among adults with schizophrenia and summarize their specific characteristics, including the concept and construct, task content, presentation modality, answer mode, control questions or items, scoring, and psychometric properties.

## 2. Materials and Methods

### 2.1. Database Search

This systematic review was conducted following the guidelines of the Preferred Reporting Items for Systematic Reviews and Meta-Analysis (PRISMA) [36]. We searched electronic databases including the Web of Science, Medline, Embase, and PsycINFO, as well as the Google Scholar search engine, up to March 2019, using the following key search terms: (“schizo” OR “psychosis”) AND (“theory of mind” OR “mentalizing”) AND (“ToM task” OR “ToM assess”).

### 2.2. Inclusion and Exclusion Criteria

The literature search was based on PICOS criteria (i.e., patients, intervention, comparison, outcomes, and study design), and in the present systematic review, P was adults with schizophrenia, I was not applicable, C was not applicable, O was ToM ability measured by ToM task, and S was not applicable. More specifically, studies were included if they: (1) reported outcomes measured by ToM tasks; (2) tested adults aged 18–60 years with schizophrenia-spectrum disorders (schizophrenia, schizoaffective, psychosis), which had been confirmed using the Structured Clinical Interview for the Diagnostic and Statistical Manual for Mental Disorders (DSM-IV, DSM-5, DSM III R) or the International Statistical Classification of Diseases and Related Health Problems (ICD-10) criteria in the analyzed studies; (3) were peer-reviewed articles; (4) were published in English or Mandarin. Studies were excluded if (1) the information for identifying the adopted ToM tasks was insufficient, (2) they were review articles, and/or (3) the full texts were not available.

### 2.3. Search Review

The search generated a total of 625 records. All the records were examined to remove 224 duplications. Two researchers independently screened the remaining 401 records by reviewing abstracts and titles. After comparing the differences in the screened results, 216 were excluded according to the presented inclusion and exclusion criteria and with the agreement of the two researchers (Y.-C.Y. and K.-L.C.). Therefore, 185 full-text versions of potential articles were retrieved. The two researchers further reviewed the full-text articles independently, and 15 more articles were obtained by checking the reference lists from the reviewed articles. To include articles in the current review, the two researchers integrated their review results and discussed the disagreements between them until a consensus was reached. Finally, a total of 117 consolidated records were included for further qualitative synthesis. The search process is outlined in Figure 1.

### 2.4. Data Extraction and Analysis

The type of ToM task was identified according to the employed concept and the method of administration. The final 117 records were qualitatively analyzed with the following extracted variables: embedded ToM concept, employed construct, number of included studies, task content, presentation modality, answer mode, inclusion of control questions or items, scoring, and psychometric properties. The psychometric properties were further evaluated using the criteria proposed in previous research. More specifically, the internal consistency, test–retest reliability, unidimensionality, convergent validity, criterion validity, and ecological validity were evaluated using the criteria proposed by the COnsensus-based Standards for the selection of health Measurement INstruments (COSMIN) [37]. Known-group validity was evaluated using the criteria proposed by Brown and Subel [38]. Internal responsiveness was evaluated using the criteria proposed by Husted et al. [39]. Table 1 summarizes all the psychometric testing with relevant criteria used in the present systematic review. To explain the trustworthiness of the results, the methodological qualities of the included studies were assessed using the COSMIN Risk of Bias Checklist [40].

## 3. Results

### 3.1. Identified ToM Tasks and Quality Assessment

After review of these 117 records, 34 ToM tasks that could be grouped into 13 types were identified. These 13 types of ToM tasks were as follows: Hinting task (HT), first-order False Belief stories (FB1), second-order False Belief stories (FB2), False Belief picture sequencing (FB-seq), Character Intention task (CIT), Visual Jokes (VJ), Irony task (IR), Faux Pas (FP), Yoni’s Verbal and Eye Gaze Cues (Yoni), Story test (ST), Movie for Social Cognition (MSC), Reading the Mind in the Eyes Test (RMET) and the “Moving Shapes” paradigm (MS).

The quality assessment of the studies that evaluated the psychometric properties (Appendix A) revealed that the methodologies or statistical approaches applied for testing the psychometric properties of specific types of tasks were doubtful due to unclear descriptions. The specific insufficiencies were as follows: the internal consistency, test–retest reliability, convergent and known-group validities of the HT; the test–retest reliability and known-group validity of the FB1; the test–retest reliability and known-group validity of the FB2; the structural and known-group validities of the FB-seq; the known-group validity of the IR; the convergent validity of the FP; the test–retest reliability of the ST; and the structural validity of the MS. The others presumably employed adequate methodologies and statistical approaches.

### 3.2. Concept and Construct

Table 2 shows the concept, construct (i.e., cognitive and affective ToM), and administration of these types of ToM tasks. Almost all the ToM tasks (11 out of 13) contained one specific ToM concept, and only the ST and MSC measured multiple concepts. The different versions of the ST measured the concepts of double bluff, mistakes, persuasion, white lie, figure of speech, lies, jokes, false beliefs, false attributions, sarcasm, and faux-pas. The various versions of the MSC contained the concepts of first- and second-order false belief, faux pas, sarcasm with metaphor, deception, humor, and persuasion.

Regarding the employed constructs, nine types of tasks measured the ToM construct specifically. The MSC, RMET, and MS measured the extra non-ToM construct, which was emotion perception [42], emotion recognition and vocabulary comprehension [43,44], and social perception and visual-spatial problem solving [45], respectively. Among the tasks measuring exclusively the ToM construct, the FB-seq and VJ required only the cognitive dimension, while the other seven types included both the cognitive and affective dimensions.

### 3.3. Administration

In general, these 13 types of tasks had diverse presentation modalities, answer modes, control questions/items, and scoring (please see Table 2). The presentation modalities in all tasks could be divided into three forms: verbal, visual, or multi-channeled. A verbal modality refers to texts or labels applied in written stories, movies, or pictures. A visual modality refers to pictures applied in ToM stories or movies. A multi-channeled modality refers to auditory (i.e., voice), verbal and visual inputs used in a video or animation. Five types of tasks (i.e., FB1, FB2, Yoni, ST, and RMET) adopted verbal and visual modalities simultaneously. Four types (i.e., FB-seq, CIT, VJ, and MS) employed visual pictures. The IR and FP used only verbal stories. The HT applied either verbal or multi-channeled ones, and the MSC used multi-channels.

Regarding the answer modes, open-ended questions were adopted in six types of tasks, including the HT, FB1, FB2, VJ, FP, and ST. The CIT, IR, Yoni, MSC, and RMET used multiple choices. The FB-seq required sequencing and description of story pictures. The MS utilized open-ended or multiple-choice questions in different versions.

To ensure the people with schizophrenia comprehended and remembered the ToM items for their further inferences of mental states, control questions or items were employed in all types of tasks, except for the HT. The FB1, FB2, IR, FP, Yoni, ST, and MSC employed control questions based on the same scenario as the ToM items and measuring basic cognition, such as memory and comprehension. Another six types utilized control items independent of the ToM items to measure the diverse types of neurocognition required to complete specific types of tasks (e.g., facial recognition as the control item in the RMET).

As for the scoring of each item, nine types, the CIT, VJ, IR, FP, Yoni, ST, MSC, RMET, and MS, applied dichotomous ratings. The HT, FB1, FB2, and ST utilized polytomous ratings. The FB-seq weighted scores according to the correctness of the order of each picture in the story. The scores of each ToM item were added up as the total score, with higher total scores indicating better ToM performance. In all, only the control questions in the FB1 and FB2 were required to be answered correctly for rating of the ToM questions to proceed.

In summary, in the reviewed articles, the ToM tasks were presented with verbal, visual, or multi-channeled modalities. To elicit responses, open-ended questions, multiple-choice questions, or sequencing and description of story pictures were used. Control questions or items were employed to clarify the influences of other cognitive confounders. Finally, the scoring could be dichotomous, polytomous, or weighted.

### 3.4. Psychometric Properties

Table 3 shows the psychometric properties (including reliability, validity, and responsiveness) of each type of ToM task. In reliability, internal consistency was assessed for the HT (ω = 0.57), FB-seq (α = 0.54), VJ (α = 0.83), FP (α = 0.816), RMET (α = 0.735), and MS (α = 0.80 to 0.84); test–retest reliability was assessed for the HT (ICC = 0.78), FB1 (ICC = 0.31), FB2 (ICC = 0.31), FP (ICC = 0.76), ST (ICC = 0.5), and RMET (ICC = 0.24, *r =* 0.753). Good internal consistencies were only reported for the VJ, FP, RMET, and MS; good test–retest reliabilities were only reported for the HT, FP, and RMET; contradictory results of test–retest reliabilities were reported for the RMET. Internal consistencies or test–retest reliabilities of the others were either poor or unexamined.

The validities were tested in all types (Table 3). Unidimensionality was investigated in the FB-seq (supported) and MS (not supported). Known-group validity was investigated in all types of tasks (*p <* 0.0001 to 0.88). Good known-group validities were found for the HT, IR, FP, ST, MSC, MS, and RMET, and for most of the FB2, CIT, and VJ, while contradictory results on the known-group validities were found for the FB1, FB-seq, and Yoni. Convergent validity was investigated in the HT (*r =* 0.352 to 0.477), FP (*r =* 0.34 to 0.68), MSC (*r =* 0.51 to 0.63), RMET (*r =* 0.46 to 0.49), and MS (*r =* 0.29 to 0.526). Criterion validity was investigated in the HT (*r =* 0.243 to 0.276), FB-seq (*r =* 0.23 to 0.31), Yoni (*r =* 0.261), ST (*r =* 0.01 to 0.24), RMET (*r =* 0.01 to 0.43), and MS (*r =* 0.23 to 0.47). Ecological validity was investigated in the ST (*r =* 0.07 to 0.19) and RMET (*r =* 0.02 to 0.26). Thus, poor convergent, criterion, or ecological validities were found for all eight investigated tasks.

Regarding the responsiveness, neither the internal responsiveness (i.e., sensitivity to change in itself) nor the external responsiveness (i.e., using external criteria) has been investigated. That is, our synthesized results indicate that the responsiveness has never been examined in any ToM task in the current literature.

## 4. Discussion

This systematic review summarized and evaluated the current ToM measures in adults with schizophrenia according to their specific characteristics, including the concept and construct, administration, and psychometric properties. In this review, 13 types of ToM tasks from 117 articles were identified for synthesis. These ToM tasks were found to vary greatly in administration (i.e., presentation modalities, answer modes, strategies of controlling cognitive confounders, and rating methods). As regards the concept and construct, most ToM tasks assess both the cognitive and affective ToM dimensions. Only the MSC, RMET, and MS assess extra non-ToM constructs. Most ToM tasks measure one specific ToM concept, and only the ST and MSC measure multiple ToM concepts. Moreover, based on the COSMIN guidelines, the psychometric evidence of the ToM tasks remains insufficient (poor or unexamined). This insufficiency indicates that ToM results should be interpreted with caution and that studies on the psychometrics of the currently existing ToM tasks are warranted.

### 4.1. Concept and Construct

Most ToM tasks measure a single ToM concept, with only the ST and MSC measuring multiple ToM concepts. Using a single specific concept may cause difficulty in measuring the whole spectrum of ToM in adults with schizophrenia. In addition, the use of diverse ToM tasks containing a single concept has resulted in inconsistent findings between studies and made their comparison or synthesis difficult [28]. Therefore, tasks with multiple concepts should be used to more comprehensively capture the whole ToM spectrum in adults with schizophrenia and compare the results among studies.

However, one of the concepts in the multiple ToM concepts of the ST and MSC, “metaphor”, has been found not to measure ToM directly. The concept of “metaphor” has been demonstrated to engage a cognitive process distinct from those of ToM [54]. Interpretation of metaphor involves a descriptive use of language, which is related to the ability to understand the logical and contextual characteristics of the concepts that are linguistically encoded. A similar ToM ability, such as recognizing irony, involves interpretation, which is related to an online ability to go beyond facts of a situation and to attribute a mental state to the speaker. Therefore, whether the concept of metaphor is ToM or not needs further discussion, and the construct validities of the ST and MSC should be further examined to ensure that all the included concepts measure the ToM construct.

Most of the ToM tasks reviewed in this study contain both dimensions, cognitive and affective ToM. The cognitive and affective dimensions assess distinct facets and are integral to ToM, as supported by a neurobiological model and studies. Abu-Ake and Shamay-Tsoory [25] concluded that both dimensions are processed by different neuropathways in the prefrontal area, with cognitive ToM engaging the dorsal stream and affective ToM engaging the ventral stream of the fronto-striatal neuronetworks. The patterns of cognitive and affective ToM impairments have been found to be different among people with schizophrenia with different presentations of psychotic symptoms, such as the violent and non-violent schizophrenia groups [63], and between people with schizophrenia and affective disorders [12]. Therefore, it is recommended that both cognitive and affective components be included in a ToM task to measure two distinct and important facets of ToM.

In addition, the MSC, RMET, and MS were found to measure extra-non-ToM constructs, which is both an advantage and a disadvantage. The MSC, RMET and MS measure multi-dimensional constructs of social cognition encompassing ToM, social perception, and emotion perception [64]. Social cognition has been reported to contribute to real-world outcomes such as social competence, community functioning, and quality of life [65]. Applying the MSC, RMET, and MS, which measure extra non-ToM constructs, may better predict the real-world functioning of people with schizophrenia. However, for assessments to identify ToM problems in people with schizophrenia, applying a task measuring exclusively the ToM construct is recommended.

### 4.2. Administration

The modalities employed by the 13 types of ToM tasks include the verbal, visual, and multi-channeled forms. Verbal or visual modalities are mostly adopted, and only the HT and MSC apply multi-channeled modalities. A multi-channeled modality, which presents task stories with verbal, visual, and auditory stimulus via videos, approximates real-life situations and is more concrete [66]. This modality may facilitate better comprehension for people with schizophrenia, who usually have cognitive impairments [58,67,68]. Therefore, a task with a multi-channeled modality is preferred for people with schizophrenia to understand task stories for answering ToM questions. However, the abilities measured by a ToM task with a multi-channeled format may include other, non-ToM concepts. For example, the Movie for the Assessment of Social Cognition (MASC) was designed with a video component to display social interactions containing various extents and qualities of language, gestures, and facial expressions. The MASC has been evidenced to measure another dimension of social cognition, the emotion perception ability, which is different from the ToM concept [58].

As for the responses, most ToM tasks adopted either open-ended or multiple-choice questions. The two different answer modes require different information retrieval processes, which lead to different sensitivities in reflecting ToM performances in people with schizophrenia. Answering an open-ended question requires a goal-directed active recollection of information from the ToM scenario (self-explanation), whereas answering a multiple-choice question relies on an automatic passive activation of pre-existing knowledge related to the scenario (familiarity). The former process, driven by an open-ended question, requires a better ability to retrieve and integrate information and then relate it to the online mental representation process, wherein the ToM scenario provides new information with less coherence with the participant’s preexisting knowledge. Therefore, instead of a multiple-choice format, we recommend using an open-ended question for responses to measure the participant’s ToM performance more precisely. However, using open-ended questions for responses could be a challenge for scoring. People with schizophrenia may respond incompletely or inappropriately, such as with short, ambiguous, or loosely associated speech. Applying follow-up clarification and probing to improve the quality of responses and to recognize the meaning of the speech of the people with schizophrenia can be a solution to this problem [69].

Control questions or items are commonly used in ToM tasks. In most of the tasks, the scores of the control question and the ToM question are rated separately. To control for confounders of other cognitive deficits on ToM, control questions or other neurocognition tests can serve as a covariate for analysis [66,70]. However, only in the FB1 and FB2 must the control questions be answered correctly for the ToM score to be rated. This requirement is to ensure that the patient’s general cognition is sufficiently good for the following inference of a mental state. Even so, asking them to access a ToM task may also exert too great a memory load on people with schizophrenia, especially when the people with schizophrenia have cognitive deficits in memory, pragmatic ability, and comprehension [68,71]. An alternative approach that has been proposed for patients with traumatic brain injury is to provide them with the story content while the control and ToM questions are asked [72,73]. Therefore, we suggest that control questions be rated independently or as the prerequisite for ToM scoring in the ToM assessment. In addition, to avoid placing excessive cognitive requirements on people with schizophrenia, the story content should be presented while they are answering the control questions.

### 4.3. Psychometric Properties

Regarding reliability, no information regarding their internal consistency and test–retest reliability has published on over half (7 out of 13) of the ToM tasks. Only about half of the rest of the ToM tasks are supported by psychometric evidence of good internal consistency and test–retest reliability. As internal consistency and test–retest reliability are important properties for understanding the stability or reproducibility of a ToM task, healthcare providers should interpret the ToM scores with caution if evidence of the reliability of the ToM task is not available.

Regarding validity, the convergent, criterion-related, or ecological validities were poor on eight types of tasks, and these validities were not investigated in the other five types. Thus, further validity evaluations on all types are needed to ensure that the concept measured by the task is highly correlated with other assessments measuring a similar concept, the same ToM construct, or real-world abilities. The known-group validities have been examined on all types. Good known-group validities were reported for most of the tasks, while inconsistent findings on known-group validities were reported for the FB1, Fb-seq, and Yoni. The less discriminating powers of the FB1, Fb-seq, and Yoni in detecting different ToM abilities between groups may be related to the employment of more basic ToM concepts, e.g., understanding another’s belief about the world contrasting with reality [27]. From another perspective, the assessments with a more basic ToM concept (e.g., FB1) detect deficits in basic ToM in people in the acute phase of schizophrenia [74]. Based on the above evidence, tasks with more basic ToM concepts may not always have great discriminating abilities for between-group comparisons. However, to identify a broader range of ToM abilities in people with schizophrenia spectrum for corresponding interventions, we suggest including the basic concept in ToM assessments. Furthermore, to responsively measure the intervention outcomes, examination of the internal and external responsiveness is also warranted.

### 4.4. Limitations and Suggestions for Further Investigations

Our evaluation and interpretation of the psychometric properties of the reviewed studies are limitations in the present systematic review. We simply evaluated and interpreted the psychometric properties using dichotomous measures (i.e., good or poor); however, psychometric properties are continuous values. A meta-analysis may be an alternative way to quantitatively synthesize study results using continuous values [75]; this may take care of the limitation we mentioned above. That is, meta-analysis can help estimate precise interpretations for each psychometric property. Unfortunately, we have found serious barriers to performing a meta-analysis, including insufficient psychometric studies and the heterogeneity of the study findings on some psychometric properties [76,77].

To address this limitation, we suggest performing an overall psychometric investigation of each type of ToM task applied in people with schizophrenia. In addition, we propose to develop a ToM measure for people with schizophrenia. This new ToM measure is expected to contain multiple concepts and both the cognitive and the affective dimension within an exclusive ToM construct, employ a multi-channel modality and open-ended questions for responses, and be assessed for its psychometric properties.

In this review, we noticed that the “ToM” concept had divergent operational definitions in the various studies and was assessed in different ways. Apart from the ToM ability being defined as “the ability to perceive others’ thoughts or feelings” in this study, the ToM ability can also be extended to the concept of “reflective functioning”. Reflective functioning (RF) (or mentalizing) refers to the ability to interpret the internal mental states, such as feelings or thoughts, of both the self and others. That is, RF involves both self- and other-mentalizing. The other-mentalizing ability has been included in the scope of our systematic review and defined as the ToM ability. In contrast, self-mentalizing involves the ability to infer ones own mental states, which is introspective, requiring inferences to be made according to one’s internal information, such as autobiographical memory and emotions. Evidence has shown that other- and self-mentalizing are dissociable, yet interactive, in a shared mentalizing neuronetwork [25]. Other-mentalizing (i.e., ToM) and self-mentalizing are different abilities and cause different responses in their respective posterior regions, including the temporoparietal junction, precuneus/posterior cingulate complex, and superior temporal sulcus in the mentalizing neuronetwork. Hence, other- and self-mentalizing have been found to be responsive to different types of ToM measures. Impairment in other-mentalizing can be detected by the ToM tasks with observable reality, which is within the scope of this systematic review. In contrast, the self-mentalizing employed in RF is more responsive to the narrative-based ToM assessments [78,79]. However, sharing the same mentalizing network, other- and self-mentalizing are interactive, providing reciprocal references for each other and contributing to RF. An individual with high RF can demonstrate some certainty about his or her own mental states and those of others, meanwhile knowing that this certainty can be modified by self-internal stored information and external information from others [80]. Given that other-mentalizing and self-mentalizing may influence each other, exploring self-mentalizing or RF in people with schizophrenia may provide a broader scope to understand their ToM difficulties.

ToM performance in the everyday contexts of people with schizophrenia has been proposed to be more complicated than the ToM ability measured by standardized ToM tasks [81]. Therefore, when interpreting the ToM impairment of people with schizophrenia in ToM tasks, several issues should be considered: (i) the causal factors of ToM deficits, e.g., the loss of neurocognitive ability and poverty of early attachment; (ii) the impact of affective states;, e.g., the intense emotions of being abused may create a bias for inferring others’ mental states; and (iii) the phase of illness in which these deficits occur and the manner in which these deficits present during the psychosocial treatment. Indeed, McCabe et al. [82] proposed a different method to assess the ToM performance of people with chronic schizophrenia. Their ToM performance was evaluated by analyzing the conversational interactions between people with schizophrenia and mental health professionals in outpatient consultations and cognitive behavioral therapy. Their results showed that the interviewee might still have persecutory delusion and at the time could correctly infer the interviewer’s intentions. That is, mentalizing in everyday interactions may be triggered and framed by the behaviors of others, unlike in controlled experimental ToM tasks.

To gain a broader understanding of the ToM performances of people with schizophrenia, we recommend that future investigations include explorations of the following: (i) an extended concept of ToM, i.e., RF (or mentalizing) and its assessment; (ii) the influence of diverse factors on ToM performances in people with schizophrenia, such as the causal factors of ToM deficits, personal emotional trauma, state of psychosis, and psychosocial interventions; and (iii) potential reliable and valid ToM measures for evaluating ToM performances in an individual’s real social context, as compared with ToM tasks, which measure clearly defined ToM concepts or ToM dimensions in experimental contexts.

## 5. Conclusions

To our knowledge, this is the first systematic review to provide an overall evaluation of the current ToM measures in schizophrenia with the rigorous PRISMA guidelines. This review provides a deeper and more comprehensive understanding of the current ToM measures in schizophrenia for mental health practitioners. Our review has identified 13 types of ToM tasks used in adults with schizophrenia and discussed their concepts and dimensions, presentation modalities, answer modes for responses, control questions and scoring, as well as the psychometric properties. Based on the results of this review, we have found several issues with the current ToM tasks in schizophrenia: measurement of a single ToM concept, inclusion of non-ToM concepts, cognitive overload due to the task scenario for people with schizophrenia, and insufficient psychometric properties. Therefore, we recommend a thorough psychometric investigation of the current ToM tasks for application to people with schizophrenia, as well as the development of new ToM tasks for people with schizophrenia as needed. We also propose principles for the present application of ToM tasks in practice, as follows: selecting a task with multiple ToM concepts, or an exclusive ToM construct containing both cognitive and affective dimensions; applying a multi-channeled modality; utilizing open-ended questions for responses; and employing control questions separately from ToM questions. Our review provides clinical and research implications in several forms: guidance for practitioners and researchers in choosing from the extant ToM tasks for people with schizophrenia; a recommendation for researchers to improve ToM tasks; and suggestions for future directions of ToM studies in people with schizophrenia.

## Figures and Tables

**Figure 1 ijerph-18-07172-f001:**
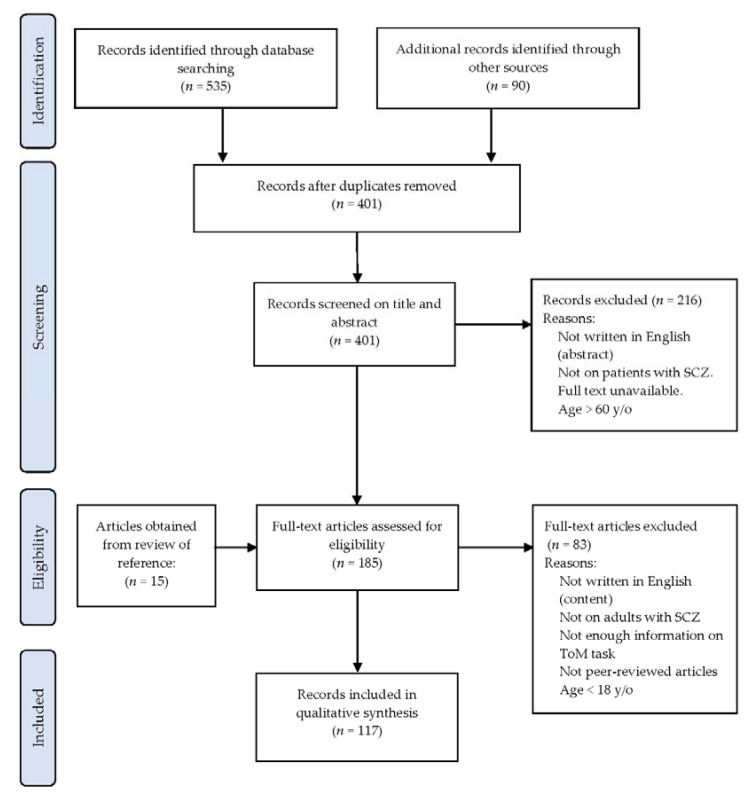
Flow diagram for records included in the systematic review. y/o = years old; SCZ = schizophrenia; ToM = theory of mind.

**Table 1 ijerph-18-07172-t001:** Criteria for evaluating the qualities of psychometric properties of current ToM tasks.

Psychometric Property	Measure	Criteria for Good Measurement Properties (Reference)
Reliability		
Internal consistency	Cronbach’s alpha (*α*) Omega (ω)	* α ≥ 0.70 ω ≥ 0.70 [41]
Test–retest reliability	Intraclass correlation coefficient (ICC) or weighted Kappa	* ICC or weighted Kappa ≥ 0.70
Construct validity		
Unidimensionality	Classical test theory Item response theory (IRT)/Rasch analysis	* CFA: CFI or TLI or comparable measure > 0.95 OR RMSEA <0.06 OR SRMR < 0.08 * No violation of unidimensionality3: CFI or TLI or comparable measure > 0.95 OR RMSEA < 0.06 OR SRMR < 0.08 AND no violation of local independence: residual correlations among the items after controlling for the dominant factor < 0.20 OR Q3′s < 0.37 AND no violation of monotonicity: adequate looking graphs OR item scalability > 0.30 AND adequate model fit: IRT: χ2 > 0.01; Rasch: infit and outfit mean squares ≥ 0.5 and ≤1.5 OR Z standardized values ≥ 2 and <2
Known-group validity	Independent *t*-test or Analysis of variance (*p*)	Significant difference: *p <* 0.05 [38]
Convergent validity	Pearson’s correlation coefficient (*r*) or Spearman’s correlation coefficient (*ρ*)	*r* or *ρ* ≥ 0.70
Criterion validity		
Concurrent, predictive	Pearson’s correlation coefficient (*r*) or Spearman’s correlation coefficient (*ρ*)	*r* or *ρ* ≥ 0.70
Ecological validities	Pearson’s correlation coefficient (*r*) or Spearman’s correlation coefficient (*ρ*)	*r* or *ρ* ≥ 0.70
Responsiveness		
Internal responsiveness	Effect size (ES) and Standardized response mean (SRM)	Low: ES and SRM = 0.2 Medium: ES and SRM = 0.5 High: ES and SRM = 0.8 [39]
External responsiveness	Area under the ROC curve (AUC)	* AUC ≥ 0.7

Note: CFA = confirmatory factor analysis; the criteria marked with * applied COSMIN [37].

**Table 2 ijerph-18-07172-t002:** Types, concept, construct and administration of the current ToM tasks.

Name of Each Type of ToM Task (Reference)		ToM Concept	Construct	*n* of Included Studies	Task Content	Presentation Modality	Answer Mode	Inclusion of Control Questions/ Items	Scoring
Hinting task V1 [33]		Infer real intentions behind indirect words	Cog and Aff	12	Ten short stories about a social interaction between two characters. Each story ends with one character dropping a hint.	Verbal stories	Open-ended questions	No	0, 1, 2.
V2 [16]					Movie sequence as presentation modality.	Multisensory movie (verbal, visual, auditory)			0, 1, 2.
First order False Belief stories V1 [46,47]	FB1	Understand one has a false belief about reality	Cog and Aff	17	Sally and Anne story; Cigarettes story.	Verbal stories and visual adds	Open-ended questions	Q: M and C	ToM: 0, 1. Explanation: 0, 1, 2, 3.
V2 [48]	FB1 with deception	Understand psychological states guide behaviors and to deceive			First order FB and deception stories with series of cartoon drawings.				0, 1, 2. Scored only when control question is correctly answered
Second order False Belief stories V1 [47,49]	FB2	Understand one has a false belief about the belief of another	Cog	15	Ice-Cream Van story; Burglar story.	Verbal stories and visual adds	Open-ended questions	Q: M and C	0, 1, 2.
V2 [48]	FB2 with deception	Understand one ignores misinformation because another is trying to deceive	Cog and Aff		Second order FB and deception stories with series of cartoon drawings.				0, 1, 2. Scored only when control question is correctly answered.
False Belief picture sequencing V1 [31]		Correctly complete a set of pictures based on false belief inferences	Cog	14	Arrange four four-card picture sequences of false beliefs in a correct order. Four types of stories: social-script, mechanical, false-belief and capture.	Visual picture sequences	Both versions: Sequence story pictures. V2 includes an additional open-ended questions	I: Inferential reasoning ability	0–6.
V2 [50]					Six picture stories of false beliefs and 23 questions with first and second order ToM and non-mental questions.				Picture sequencing: 0–6; ToM questionnaire: 0, 1.
Character Intention task [51,52]		Understand the intention of a person in subtle social cues	Cog and Aff	5	Thirty or 42 sets of comic strips. Each strip: Three pictures in sequence and answer cards (Attribution of intention and Attribution of false belief).	Visual pictures	Multiple choice questions	I: Basic reasoning ability	0, 1.
Visual Jokes [53]		Detect visual jokes involving attribution of ignorance, false belief or deception	Cog	5	Two sets of 10 cartoon jokes. Set 1: Mental state attribution to false belief and deception. Set 2: Physical/behavior scene.	Visual jokes	Open-ended questions	I: Other general cognitive deficits	0, 1.
Irony task [34,54]		Understand the opposition between literal and true meanings of words	Cog and Aff	5	Nine or more stories with ironical utterance.	Verbal stories and written copy	Multiple choice questions	Q: C	0, 1.
Faux Pas [30,55]		Infer different perspectives: speaker’s thinking and listener’s feeling	Cog and Aff	11	FP stories with questions of recognition and understanding of FP.	Verbal stories with a print copy	Open-ended questions	I: Basic reasoning ability, attention, Q: M or C.	0, 1.
Yoni’s Verbal and Eye Gaze Cues [26]		Judge mental states based on verbal and eye gaze cues.	Cog and Aff	6	Each of 87 trials: a cartoon outline of a face and four colored pictures around each corner. Questions: first-order or second-order ToM, cognitive or affective ToM.	Visual and Verbal (written questions)	Multiple choice questions	Q: Attention and C	0, 1.
Story test V1 [35]		Multiple concepts: various, 4–5	Cog and Aff	6	All three versions use stories. Eight ToM: Double bluff, mistakes, persuasion, white lies.	Verbal and visual adds	Open-ended questions	I: Cause-effect inference	0, 1.
V2 [56]					Five ToM: Figure of speech, lies, white lies, joke.			Q: M and C	0, 1.
V3 [57]					Eighteen ToM stories. False beliefs, false attributions, lies, sarcasm, faux pas.			Q: M and C	0, 1, 2.
Movie for Social Cognition V1 [58]		Multiple concepts: 5	Cog and Aff ToM; Emotion perception	3	Movie for the Assessment of Social Cognition (MASC): 15 min movie about characters getting together for a dinner party: paused 46 times for 48 questions. ToM: first- and second-order false belief, faux pas, metaphor, or sarcasm.	Scenario: multi-modalities. Question: verbal and written	Multiple-choice questions	Q: M and C, I: Reasoning ability	0, 1. Outputs: Error categories, mental state modalities and non-social inferencing. M and C: 1, 0.5, 0.
V2 [59]		Multiple concepts: false belief, deception, faux pas, humor, sarcasm, and persuasion			Virtual Assessment of Mentalising Ability (VAMA): 12 video clips depicting a social drama within interactive virtual environment.	Interactive multi-modalities		I: Reasoning ability	Scored in two ways. Three-point scale: 0 (impaired), 0 (hyper), 1 (reduced) and 2 (accurate). Dichotomous scale: 1 (accurate), 0 (incorrect: any wrong response).
Reading the Mind in the Eyes Test [32]		Infer mental states from the pictures of persons’ eyes and apply affective terms	Emotion recognition; Vocabulary comprehension; Aff ToM	9	Thirty-six eye photos showing emotions. Choose one term from four choices. Include definitions of emotional terms for reference.	Visual photos and verbal question	Multiple-choice questions	I: Face-recognition problems	0, 1
The “Moving Shapes” paradigm V1 [60]		Infer intentions of silent cartoon figures enacting social drama	ToM, Social perception; Visual-spatial Problem solving	6	Twelve animations with two characters, a big red triangle and a small blue triangle, moving on framed white background.	Non-verbal animations	Open-ended questions	I: Alexithymia problem and empathetic ability	Four dimensions: Intentionality: 0–5; Appropriateness: 0–3. Certainty: 0–3; Length: 0–4.
V2 [60,61,62]					A large triangle, small triangle and small circle enact social drama. Both versions contain following questions.		Multiple choice questions	No	0, 1.

Note: *n* = numbers; Cog = Cognitive; Aff = Affective; V1 = Version 1; V2 = Version 2; Q = Question; I = Items; M = Memory; C = Comprehension; First order False belief = FB1; Second order False belief = FB2.

**Table 3 ijerph-18-07172-t003:** Psychometric properties of each type of ToM task.

ToM Task	*n* of Included Studies	Reliability (*n* of Reporting Study)		Validity (*n* of Reporting Study)				
		Internal Consistency	Test–retest Reliability	Unidimensionality	Known-Group Validity (SCZ vs. HC)	Convergent Validity	Criterion Validity	Ecological Validity
HT	11	ω = 0.57 (1)	ICC = 0.78 (1)	NA	*p <* 0.0001 to *p* = 0.03 (11)	*r* = 0.352 to 0.477 (5)	*r* = 0.243 to 0.276 (2)	NA
FB1	15	NA	ICC = 0.31 (1)	NA	*p =* 0.055 to 0.293 (7); *p <* 0.001 to *p <* 0.01 (7)	NA	NA	NA
FB2	14	NA	ICC = 0.31 (1)	NA	*p =* 0.17 to 0.27 (3); *p* < 0.0001 to *p =* 0.02 (11)	NA	NA	NA
FB-seq	14	*α* = 0.54	NA	Supported: IRT: *χ*2 (2) = 3.65, *p* = 0.186, CFI = 0.994, TLI = 0.988, RMSEA = 0.054 (1)	*p* = 0.056 to 0.282 (3); *p* < 0.0005 to *p* = 0.023 (9)	NA	*r =* 0.23 to 0.31 (1)	NA
CIT	4	NA	NA	NA	*p* = 0.88 (1); *p* < 0.0001 to *p* < 0.05 (3)	NA	NA	NA
VJ	5	*α* = 0.83 (1)	NA	NA	*p* = 0.08 (1); *p* < 0.0001 to *p* < 0.001 (4)	NA	NA	NA
IR	5	NA	NA	NA	*p* < 0.0001 to *p* < 0.01 (5)	NA	NA	NA
FP	11	α = 0.816 (1)	ICC = 0.76 (1)	NA	*p =* 0.0003 to 0.041 (9)	*r =* 0.34 to 0.68 (3)	NA	NA
Yoni	6	NA	NA	NA	*p* > 0.05 on first order ToM (2); *p >* 0.05 on cognitive ToM (2); *p* < 0.001 to *p* = 0.049 (6)	NA	*r =* -0.261 (1)	NA
ST	5	NA	ICC = 0.5 (1)	NA	*p <* 0.001 to 0.038 (4)	NA	*r = 0*.01 to 0.24 (1)	*r = 0*.07 to 0.19 (1)
MSC	3	NA	NA	NA	*p <* 0.001 (3)	*r =* 0.51 to 0.63 (3)	NA	NA
RMET	9	α = 0.735 (1)	ICC = 0.24 (1); *r* = 0.753 (1)	NA	*p <* 0.0001 to *p* < 0.05 (8)	*r =* 0.46 to 0.49 (1)	*r =* 0.01 to 0.43 (2)	*r =* 0.02 to 0.26 (1)
MS	6	α = 0.80 to 0.84 (2)	NA	Not supported: *χ*2 (152) = 194.997, TLI = 0.858, CFI = 0.873, RMSEA = 0.069 (1)	*p <* 0.0001 to *p =* 0.001 (5)	*r =* 0.29 to 0.526 (4)	*r =* 0.29 to 0.47 (5); *r =* 0.23 (1)	NA

Note: *n* = numbers; criterion = criterion-related; NA = no information from the analyzed studies; SCZ = schizophrenia; HC = healthy control; no studies have reported the responsiveness. External criteria of social functioning, independent living skills, or psychotic symptoms were used for convergent validity; other measures of ToM for criterion validity; and self-reported real-life social functioning for ecological validity.

## Data Availability

Data are contained within the article or Appendix A.

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
