# Peer review of "A Systematic Review of the Current Measures of Theory of Mind in Adults with Schizophrenia"

_ijerph, 2021, doi:10.3390/ijerph18137172_

Round 1

Reviewer 1 Report

The review “A Systematic Review of the Current Measures of Theory of 2 Mind in Adults with Schizophrenia” explores a relevant topic in the field of schizophrenia, which is the evaluation of ToM. This paper evaluated different ToM tasks, by showing the different modalities used in the research fiel to address to topic.

The article is interesting. However, some minor aspects would need to be revised:

- In the introduction a short review of the networks associated to cognitive an d affective ToM should be given, to make the manuscript more complete

- In the introduction authors should cite the others diseases in which a ToM impairment is found (such as ASD)

- Tables should be re-formatted. Right now they are difficult to understand

- Could the authors be more specific suggesting some practical implications of their study? Specific and practical information would be very welcomed to add value to their conclusions.

Author Response

Response to Reviewer’s comments

Reviewer 1

The review “A Systematic Review of the Current Measures of Theory of Mind in Adults with Schizophrenia” explores a relevant topic in the field of schizophrenia, which is the evaluation of ToM. This paper evaluated different ToM tasks, by showing the different modalities used in the research field to address to topic.

The article is interesting. However, some minor aspects would need to be revised:

  1. In the introduction a short review of the networks associated to cognitive an affective ToM should be given, to make the manuscript more complete

Response:

Thank you for the suggestion. We have added a short review of the networks associated with cognitive and affective ToM in the Introduction, as follows.

“Abu-Akel & Shamay-Tsoory [25] have concluded that cognitive ToM is mediated by the dorsal stream of the ToM neuro-networks, such as the dorsomedial prefrontal cortex, dorsal anterior cingulate cortex and dorsal striatum; affective ToM, by the ventral stream, such as the ventromedial and orbitofrontal cortices, ventral anterior cingulate cortex, amygdala and ventral striatum.” (Page 2, Lines 64–69)

Reference:

  1. Abu-Akel, A.; Shamay-Tsoory, S. Neuroanatomical and neurochemical bases of theory of mind. Neuropsychologia. 2011, 49, 2971-2984..

  1. In the introduction authors should cite the other diseases in which a ToM impairment is found (such as ASD)

Response:

Thank you for the suggestion. We have cited the references of other diseases in which ToM impairment is found and added related information in the Introduction, as follows.

“Impairments of ToM have been reported in several mental illnesses, such as schizophrenia, autism spectrum disorders, major depression, bipolar disorder, borderline personality disorder, and Parkinson’s disease [3-7]. However, the ToM performance in people with schizophrenia reveals specific features that are distinct from those in other illnesses.” (Pages 1–2, Lines 46–50)

References:

  1. Chung, Y.S.; Barch, D.; Strube,M. A meta-analysis of mentalizing impairments in adults with schizophrenia and autism spectrum disorder. Schizophr. Bull. 2014, 40, 602-616.
  2. Bora, E.; Walterfang, M.; Velakoulis, D. Theory of mind in behavioural-variant frontotemporal dementia and alzheimer's disease: A meta-analysis. J. Neurol. Neurosurg. Psychiatry. 2015, 86, 714-719.
  3. Bora, E.; Bartholomeusz, C.; Pantelis, C. Meta-analysis of theory of mind (ToM) impairment in bipolar disorder. Psychol. Med. 2016, 46, 253-264.
  4. Bora, E.; Berk, M. Theory of mind in major depressive disorder: A meta-analysis. J. Affect. Disord. 2016, 191, 49-55.
  5. Nemeth, N.; Matrai, P.; Hegyi, P.; Czeh, B.; Czopf, L.; Hussain, A.; Pammer, J.; Szabo, I.; Solymar, M.; Kiss, L.; Hartmann, P.; Szilagyi, A.L.; Kiss, Z.; Simon, M. Theory of mind disturbances in borderline personality disorder: A meta-analysis. Psychiatry Res. 2018, 270, 143-153.

  1. Tables should be re-formatted. Right now they are difficult to understand

Response:

Thank you for the suggestion. We have re-formatted Table 1, Table 2, and Table 3, as follows. (Pages 6, 9–12, 18–19)      

Table 1. Criteria for evaluating the qualities of psychometric properties of current ToM tasks

Psychometric property

Measure

Criteria for good measurement properties (reference)

Reliability

Internal consistency

Cronbach’s alpha (α)

Omega (ω)

*α ≥ 0.70

ω ≥ 0.70 [42]

Test-retest reliability

Intraclass correlation coefficient (ICC)

or weighted Kappa

*ICC or weighted Kappa ≥ 0.70

Construct validity

Unidimensionality

Classical test theory

Item response theory (IRT)/ Rasch analysis

*CFA: CFI or TLI or comparable measure > 0.95 OR RMSEA

< 0.06 OR SRMR < 0.08

*No violation of unidimensionality3: CFI or TLI or comparable measure > 0.95 OR RMSEA < 0.06 OR SRMR < 0.08 AND no violation of local independence: residual correlations among the items after controlling for the dominant factor < 0.20 OR Q3's < 0.37 AND no violation of monotonicity: adequate looking graphs OR item scalability > 0.30 AND adequate model fit: IRT: χ2 > 0.01; Rasch: infit and outfit mean squares ≥ 0.5 and ≤ 1.5 OR Z standardized values ≥ 2 and < 2

Known-group validity

Independent t-test or Analysis of variance (p)

Significant difference: p < 0.05

[39]

Convergent validity

Pearson's correlation coefficient (r) or Spearman's correlation coefficient (ρ)

r or ρ ≥ 0.70

Criterion validity

Concurrent, predictive

Pearson's correlation coefficient (r) or Spearman's correlation coefficient (ρ)

r or ρ ≥ 0.70

Ecological validities

Pearson's correlation coefficient (r) or

Spearman's correlation coefficient (ρ)

r or ρ ≥ 0.70

Responsiveness

Internal responsiveness

Effect size (ES) & Standardized response mean (SRM)

Low: ES & SRM = 0.2

Medium: ES & SRM = 0.5

High: ES & SRM = 0.8

[40]

External responsiveness

Area under the ROC curve (AUC)

*AUC ≥ 0.7

Note. CFA = confirmatory factor analysis; the criteria marked with * applied COSMIN.

Table 2. Types, Concept, Construct and Administration of the Current ToM Tasks.

Name of Each Type of ToM Task (Reference)

ToM Concept

Construct

N of Included Studies

Task Content

Presentation Modality

Answer Mode

Inclusion of Control Questions/

Items

Scoring

Hinting task

V1 [47]

Infer real intentions behind indirect words

Cog & Aff

12

10 short stories about a social interaction between two characters. Each story ends with one character dropping a hint.

Verbal stories

Open-ended questions

No

0, 1, 2.

V2 [16]

Movie sequence as presentation modality.

Multisensory movie (verbal, visual, auditory)

0, 1, 2.

First order False Belief stories

V1 [48, 49]

FB1

Understand one has a false belief about the reality

Cog & Aff

17

Sally and Anne story & Cigarettes story

Verbal stories & visual adds

Open-ended questions

Q: M & C

ToM: 0, 1. Explanation: 0, 1, 2, 3. 

V2 [50]

-FB1 with deception

Understand psychological states guide behaviors & to deceive

Cog & Aff

First order FB and deception stories with series of cartoon drawings.

0, 1, 2.

Scored only when control question is correctly answered

Second order False Belief stories

V1 [49, 51]

FB2

Understand one has a false belief about the belief of another one

Cog

15

Ice-Cream Van story; Burglar story.

Verbal stories & visual adds

Open-ended questions

Q: M & C

0, 1, 2. 

V2 [50]

-FB2 with deception

Understand one ignores misinformation because another one is trying to deceive

Cog & Aff

Second order FB and deception stories with series of cartoon drawings

0, 1, 2.

Scored only when control question is correctly answered.

Table 2. (Continued)

Name of Each Type of ToM Task (Reference)

ToM Concept

Construct

N of Included Studies

Content of Task

Presentation Modality

Answer Mode

Inclusion of Control Questions/Items

Scoring

False Belief picture sequencing

V1 [31]

Correctly complete a set of pictures based on false belief inferences

Cog

14

Arrange 4 four-card picture sequences of false beliefs in a correct order.

4 types of stories: social-script, mechanical, false-belief & capture.

Visual picture sequences

Both versions: Sequence story pictures.

V2 includes an additional open-ended questions

I: Inferential reasoning ability 

0-6.

V2 [52]

6 picture stories of false beliefs & 23 questions with first & second order ToM and non-mental questions.

Picture sequencing: 0-6;

ToM questionnaire: 0, 1.

Character Intention task [53, 54]

Understand the intention of a person in subtle social cues

Cog & Aff

5

30 or 42 sets of comic strips. Each strip: 3 pictures in sequence & answer cards (Attribution of intention & Attribution of false belief).

Visual pictures

Multiple choice questions

I: Basic reasoning ability

0, 1.

Visual Jokes [55]

Detect visual jokes involving attribution of ignorance, false belief or deception

Cog

5

Two sets of 10 cartoon jokes. Set 1: Mental state attribution to false belief and deception. Set 2: Physical/behavior scene.

Visual jokes

Open-ended questions

I: Other general cognitive deficits

0, 1.

Irony task [34, 56]

Understand the opposition between literal & true meanings of words

Cog & Aff

5

9 or more stories with ironical utterance.

Verbal stories & written copy

Multiple choice questions

Q: C   

0, 1.

Faux Pas [30, 57]

Infer different perspectives: speaker’s thinking & listener’s feeling

Cog & Aff

11

FP stories with questions of recognition & understanding of FP.

Verbal stories with a print copy

Open-ended questions

I: Basic reasoning ability, attention, Q: M or C. 

0, 1.

Table 2. (Continued)

Name of Each Type of ToM Task (Reference)

ToM Concept

Construct

N of Included Studies

Content of Task

Presentation Modality

Answer Mode

Inclusion of Control Questions/Items

Scoring

Yoni's Verbal and Eye Gaze Cues [26]

Judge mental states based on verbal and eye gaze cues.

Cog & Aff

6

Each of 87 trials: a cartoon outline of a face and four colored pictures around each corner.

Questions: first-order or second-order ToM, cognitive or affective ToM.

Visual & Verbal (written questions)

Multiple choice questions

Q: Attention & C

0, 1.

Story test

V1 [35]

Multiple concepts: various, 4-5

Cog & Aff

6

All 3 versions use stories.

8 ToM: Double bluff, mistakes, persuasion, white lies.

Verbal & visual adds

Open-ended questions. 

I: Cause-effect inference

0, 1.

V2 [58]

5 ToM: Figure of speech, lies, white lies, joke.

Q: M & C

0, 1.

V3 [59]   

18 ToM stories. False beliefs, false attributions, lies, sarcasm, faux pas.  

Q: M & C

0, 1, 2.

Movie for Social Cognition

V1 [60]

Multiple concepts: 5.

Cog & Aff ToM;

Emotion perception

3

Movie for the Assessment of Social Cognition (MASC): 15 min movie about characters getting together for a dinner party: paused 46 times for 48 questions. ToM: first- and second-order false belief, faux pas, metaphor, or sarcasm.

Scenario: multi-modalities.

Question: verbal & written

Multiple-choice questions

Q: M & C,      I: Reasoning ability

0, 1. Outputs: Error categories, mental state modalities and non-social inferencing.

M & C: 1, 0.5, 0.

V2 [61]

Multiple concepts: false belief, deception, faux pas, humor, sarcasm, and persuasion

Cog & Aff ToM;

Emotion perception

Virtual Assessment of Mentalising Ability (VAMA): 12 video clips depicting a social drama within interactive virtual environment.

Interactive multi-modalities

Multiple-choice questions

I: Reasoning ability

Scored in two ways.

3-point scale: 0 (impaired), 0 (hyper) , 1 (reduced) & 2 (accurate).

Dichotomous scale: 1 (accurate), 0 (incorrect: any wrong response).

Table 2. (Continued)

Name of Each Type of ToM Task (Reference)

ToM Concept

Construct

N of Included Studies

Content of Task

Presentation Modality

Answer Mode

Inclusion of Control Questions/Items

Scoring

Reading the Mind in the Eyes Test [32]

Infer mental states from the pictures of persons’ eyes & apply affective terms

Emotion recognition; Vocabulary comprehension; Aff ToM

9

36 eye photos showing emotions. Choose one term from 4 choices. Include definitions of emotional terms for reference.

Visual photos & Verbal question

Multiple-choice questions

I: Face-recognition problems

0, 1

The "Moving Shapes" paradigm

V1 [62]

Infer intentions of silent cartoon figures enacting social drama

ToM, Social perception; Visual-spatial Problem solving

6

12 animations with two characters, a big red triangle and a small blue triangle, moving on framed white background.

Non-verbal animations

Open-ended questions

I: Alexithymia problem and empathetic ability 

4 dimensions: Intentionality: 0-5;

Appropriateness: 0-3.

Certainty: 0-3;

Length: 0-4.

V2 [63, 64]

a large triangle, small triangle and small circle enact social drama.

Both versions contain following questions.

Multiple choice questions

No

0, 1.

Note: N = numbers; Cog = Cognitive; Aff = Affective; V1 = Version 1; V2 = Version 2; Q = Question; I = Items; M = Memory; C = Comprehension; First order False belief = FB1; Second order False belief = FB2.

Table 3. Psychometric properties of each type of ToM task.

ToM task

N of included studies

Reliability (N of reporting study)

Validity (N of reporting study)

Internal consistency

Test-retest reliability

Unidimensionality

Known-group validity (SCZ vs. HC)

Convergent validity

Criterion validity

Ecological validity

HT

11

ω = 0.57(1)

ICC = 0.78 (1)

NA

p < 0.0001 to p = 0.03 (11)

r = 0.352 to 0.477 (5)

r = 0.243 to 0.276 (2)

NA

FB1

15

NA

ICC = 0.31 (1)

NA

p = 0.055 to 0.293 (7);

p < 0.001 to p < 0.01 (7)

NA

NA

NA

FB2

14

NA

ICC = 0.31 (1)

NA

p = 0.17 to 0.27 (3);

p < 0.0001 to p = 0.02 (11)

NA

NA

NA

FB-seq

14

α = 0.54

NA

Supported: IRT: χ2 (2) = 3.65, p = .186, CFI = 0.994, TLI = 0.988, RMSEA = 0.054 (1)

p = 0.056 to 0.282 (3);

p < 0.0005 to p = 0.023 (9)

NA

r = 0.23 to 0.31 (1)

NA

CIT

4

NA

NA

NA

p = 0.88 (1);

p < 0.0001 to  < 0.05 (3)

NA

NA

NA

VJ

5

α = 0.83 (1)

NA

NA

p = 0.08 (1);

p < 0.0001 to  < 0.001 (4)

NA

NA

NA

IR

5

NA

NA

NA

p < 0.0001 to  < 0.01 (5)

NA

NA

NA

FP

11

α = 0.816 (1)

ICC = 0.76 (1)

NA

p = 0.0003 to 0.041 (9)

r = 0.34 to 0.68 (3)

NA

NA

Table 3. (Continued)

ToM task

N of included studies

Reliability (N of reporting study)

Validity (N of reporting study)

Internal consistency

Test-retest reliability

Unidimensionality

Known-group validity (SCZ vs. HC)

Convergent validity

Criterion validity

Ecological validity

Yoni

6

NA

NA

NA

p >0.05 on first order ToM (2); p >0.05 on cognitive ToM (2);

p < 0.001 to p = 0.049 (6)

NA

r = -0.261 (1)

NA

ST

5

NA

ICC = 0.5 (1)

NA

p < 0.001 to 0.038 (4)

NA

r = 0.01 to 0.24 (1)

r = 0.07 to 0.19 (1)

MSC

3

NA

NA

NA

p < 0.001 (3)

r = 0.51 to 0.63 (3)

NA

NA

RMET

9

α = 0.735 (1)

ICC = 0.24 (1);

r = 0.753 (1)

NA

p < 0.0001 to p < 0.05 (8)

r = 0.46 to 0.49 (1)

r = 0.01 to 0.43 (2)

r = 0.02 to 0.26 (1)

MS

6

α = 0.80 to 0.84 (2)

NA

Not supported:

χ2 (152) = 194.997, TLI = 0.858, CFI = 0.873, RMSEA = 0.069 (1)

p < 0.0001 to p = 0.001 (5)

r = 0.29 to 0.526 (4)

r = 0.29 to 0.47 (5);

r = 0.23 (1)

NA

Note: N = numbers; criterion = criterion-related; NA = no information from the analyzed studies; SCZ = schizophrenia; HC = healthy control; No studies have reported the responsiveness. External criteria of social functioning, independent living skills, or psychotic symptoms were used for convergent validity; other measures of ToM for criterion validity; and self-reported real-life social functioning for ecological validity.

  1. Could the authors be more specific suggesting some practical implications of their study? Specific and practical information would be very welcomed to add value to their conclusions

Response:

Thank you for the suggestion. We have added more information on the implications of our study in the Conclusions, as follows.

“To our knowledge, this is the first systematic review to provide an overall evaluation of the current ToM measures in schizophrenia with the rigorous PRISMA guidelines. This review provides a deeper and more comprehensive understanding of the current ToM measures in schizophrenia for mental health practitioners. Our review has identified 13 types of ToM tasks used in adults with schizophrenia and discussed their concepts and dimensions, presentation modalities, answer modes for responses, control questions and scoring, as well as the psychometric properties. Based on the results of this review, we have found several issues in the current ToM tasks in schizophrenia: measurement of a single ToM concept, inclusion of non-ToM concepts, cognitive overload due to the task scenario for people with schizophrenia, and insufficient psychometric properties. Therefore, we recommend a thorough psychometric investigation of the current ToM tasks for application to people with schizophrenia, as well as the development of new ToM tasks for people with schizophrenia as needed. We also propose principles for the present application of ToM tasks in practice, as follows: selecting a task with multiple ToM concepts, or an exclusive ToM construct containing both cognitive and affective dimensions; applying a multi-channeled modality; utilizing open-ended questions for responses; and employing control questions separately from ToM questions. Our review provides clinical and research implications in several forms: guidance for practitioners and researchers in choosing from the extant ToM tasks for people with schizophrenia; a recommendation for researchers to improve ToM tasks; and suggestions for future directions of ToM studies in people with schizophrenia.” (Pages 24–25, Lines 559–580)

Reviewer 2 Report

Thank you for the manuscript, please could you include also recent application studies of ToM assessment tools (e.g., Steinmair, D., et al. (2021). Mind reading improvements by mentalization based therapy training. The Bulletin of the Menninger Clinic, 85(1), 59-83.)

The strength of the manuscript lies in the concise application of guidelines (Prisma) when conducting a systematic research, especially the COSMIN criteria add value! The PICOS criteria could be added, as the 2.2. inclusion and exclusion section describes that. The limitation lies in the lack of multi-channel assessment tools. Further additional comments to the authors:

As the results section the topic "3.2. concept and construct" is reported, this should also be elaborated in the introduction in more detail. The authors only speak of ToM, but neglect mentalization/reflective functioning/social cognition literature and how to include or merge or differentiate these concepts.

The lack of multi-channel-assessment tools should be addressed in more detail, further research directions should be elaborated (please remove the paragraph line 400-403, as this is only some kind of general sentence!). 

Further readings that could be helpful and should be included in the introduction or discussion are:

Dziobek, I., Fleck, S., Kalbe, E., Rogers, K., Hassenstab, J., Brand, M.,
Kessler, J., Woike, J. K., Wolf, O. T., & Convit, A. (2006). Introducing
MASC: A movie for the assessment of social cognition. Journal
of Autism and Developmental Disorders, 36(5), 623–636. https://doi.org/10.1007/s10803-006-0107-0

Dimaggio, G., Popolo, R., Salvatore, G., & Lysaker, P. H. (2013).
Mentalizing in schizophrenia is more than just solving theory of
mind tasks. Frontiers in Psychology, 4, 83. https://doi.org/10.3389/fpsyg.2013.00083

Fonagy, P., Luyten, P., Moulton-Perkins, A., Lee, Y.-W., Warren, F., Howard,
S., Ghinai, R., Fearon, P., & Lowyck, B. (2016). Development and validation of a self-report measure of mentalizing: The reflective functioning questionnaire. PLoS One, 11(7), Article e0158678. https://doi.org/10.1371/journal.pone.0158678

Richter, F., Steinmair, D., Löffler-Stastka, H. (2021). Construct Validity of the Mentalization Scale (MentS) within a mixed psychiatric sample. Frontiers in Psychology, 12:608214. doi: 10.3389/fpsyg.2021.608214.

Steinmair, D., Horn, R., Richter, F., Wong, G., Löffler-Stastka, H. (2021). Mind reading improvements by mentalization based therapy training. The Bulletin of the Menninger Clinic, 85(1), 59-83.

Further please add for further outlook the necessary discussion to distinguish or merge ToM and Mentalization/Reflecting functioning and add also add clinical application studies in your argumentation. 

Author Response

Response to Reviewer’s comments

Reviewer 2

1.      Thank you for the manuscript, please could you include also recent application studies of ToM assessment tools (e.g., Steinmair, D., et al. (2021). Mind reading improvements by mentalization based therapy training. The Bulletin of the Menninger Clinic, 85(1), 59-83.) Response:Thank you for the suggestion. We have added recent application studies of ToM assessment tools in the Introduction, as follows.“Fortunately, it has been evidenced that the ToM impairment in people with schizophrenia can be improved by mentalizing-based interventions [19-21]. Measuring ToM in people with schizophrenia is crucial because ToM has been identified as an important target and outcome measure for psychosocial treatment.” (Page 2, Lines 55-59)References:

  1. Vass, E.; Fekete, Z.; Simon, V.; Simon, L. Interventions for the treatment of theory of mind deficits in schizophrenia: Systematic literature review. Psychiatry Res. 2018, 267, 37-47.
  2. Vass, E.; Simon, V.; Fekete, Z.; Lencse, L.; Ecseri, M.; Kis, B.; Simon, L. A novel virtual reality-based theory of mind intervention for outpatients with schizophrenia: A proof-of-concept pilot study. Clin. Psychol. Psychother. 2020, 1-12.
  3. Steinmair, D.; Horn, R.; Richter, F.; Wong, G.; Löffler Stastka, H. Mind reading improvements in mentalization-based therapy training. Bull. Menninger Clin. 2021, 85, 59-82.

  1. The strength of the manuscript lies in the concise application of guidelines (Prisma) when conducting a systematic research, especially the COSMIN criteria add value! The PICOS criteria could be added, as the 2.2. inclusion and exclusion section describes that. The limitation lies in the lack of multi-channel assessment tools. Further additional comments to the authors:

Response:Thank you for the suggestion. We have added the PICOS criteria to describe the inclusion and exclusion criteria in section 2.2, as follows.“The literature search was based on PICOS criteria (i.e., patients, intervention, comparison, outcomes, and study design), and in the present systematic review, P was adults with schizophrenia, I was not applicable, C was not applicable, O was ToM ability measured by ToM task, and S was not applicable. More specifically, …” (Page 3, Lines 124–127) 3.      As the results section the topic "3.2. concept and construct" is reported, this should also be elaborated in the introduction in more detail. The authors only speak of ToM, but neglect mentalization/reflective functioning/social cognition literature and how to include or merge or differentiate these concepts.Response:Thank you for noting this issue. We have read the articles which you recommended [81-82, 83] and added an explanation of this issue in the section 4.4 in Discussion, as follows.  “In this review, we noticed that the “ToM” concept had divergent operational definitions in the various studies and was assessed in different ways. Apart from the ToM ability being defined as “the ability to perceive others’ thoughts or feelings” in this study, the ToM ability can also be extended to the concept of “reflective functioning”. Reflective functioning (RF) (or mentalizing) refers to the ability to interpret the internal mental states, such as feelings or thoughts, of both the self and others. That is, RF involves both self- and other-mentalizing. The other-mentalizing ability has been included in the scope of our systematic review and defined as the ToM ability. In contrast, self-mentalizing involves the ability to infer self-mental states, which is introspective. It is inferring according to one’s internal information, such as autobiographical memory and emotions. Evidence has shown that other- and self-mentalizing are dissociable, yet interactive, in a shared mentalizing neuro-network [25]. Other-mentalizing (i.e., ToM) and self-mentalizing are different abilities and cause different responses in their respective posterior regions, including the temporoparietal junction, precuneus/posterior cingulate complex, and superior temporal sulcus in the mentalizing neuro-network. Hence, other- and self-mentalizing have been found to be responsive to different types of ToM measures. Impairment in other-mentalizing can be detected by the ToM tasks with observable reality, which is within the scope of this systematic review. In contrast, the self-mentalizing employed in RF is more responsive to the narrative-based ToM assessments [80, 81]. However, sharing the same mentalizing network, other- and self-mentalizing are interactive, providing reciprocal references for each other and contributing to RF. An individual with high RF can demonstrate some certainty about his or her own mental states and those of others, meanwhile knowing that this certainty can be modified by self-internal stored information and external information from others [82]. Given that other-mentalizing and self-mentalizing may influence each other, exploring self-mentalizing or RF in people with schizophrenia may provide a broader scope to understand their ToM difficulties.” (Pages 23–24, Lines 504–529)References:

  1. Fertuck, E.A.; Mergenthaler, E.; Target, M.;  Levy, K.N.; Clarkin, J.F. Development and criterion validity of a computerized text analysis measure of reflective functioning. Psychother. Res. 2012, 22, 298-305.
  2. Richter, F.; Steinmair, D.; Löffler-Stastka, H. Construct validity of the mentalization scale (MentS) within a mixed psychiatric sample. Front. Psychol. 2021, 12.
  3. Fonagy, P.; Luyten, P.; Moulton-Perkins, A.; Lee, Y.-W.; Warren, F.; Howard, S.; Ghinai, R.; Fearon, P.; Lowyck, B. Development and validation of a self-report measure of mentalizing: The reflective functioning questionnaire. PLoS One. 2016, 11, e0158678.
  4. Dimaggio, G.; Popolo, R.; Salvatore, G.; Lysaker, P. Mentalizing in schizophrenia is more than just solving theory of mind tasks. Front. Psychol. 2013, 4, 83.
  5. McCabe, R.; Leudar, I.; Antaki, C. Do people with schizophrenia display theory of mind deficits in clinical interactions? Psychol. Med. 2004, 34, 401-412.

 4.      The lack of multi-channel-assessment tools should be addressed in more detail, further research directions should be elaborated (please remove the paragraph line 400-403, as this is only some kind of general sentence!).Response:Thank you for the suggestion. We have added a more detailed description on the multi-channel-assessment tools in the section 4.2. Administration, as follows.“A multi-channeled modality, which presents task stories with verbal, visual, and auditory stimulus via videos, approximates real-life situations and is more concrete to be understood [68]. This modality may facilitate better comprehension for people with schizophrenia, who usually have cognitive impairments [60, 69, 70]. Therefore, a task with a multi-channeled modality is preferred for people with schizophrenia to understand task stories for answering ToM questions. However, the abilities measured by a ToM task with a multi-channeled format may include other, non-ToM concepts. For example, the Movie for the Assessment of Social Cognition (MASC) was designed with a video component to display social interactions containing various extents and qualities of language, gestures, and facial expressions. The MASC has been evidenced to measure another dimension of social cognition, the emotion perception ability, which is different from the ToM concept.” (Page 22, Lines 419–425)References:

  1. Dziobek, I.; Fleck, S.; Kalbe, E.; Rogers, K.; Hassenstab, J.; Brand, M.; Kessler, J.; Woike, J.; Wolf, O.; Convit, A. Introducing MASC: A movie for the assessment of social cognition. J. Autism Dev. Disord. 2006, 36, 623-636.

 We have also elaborated further research directions in the Discussion and Conclusions, as follows. In Discussion: “To gain a broader understanding of the ToM performances of people with schizophrenia, we recommend that future investigations include explorations of the following: (i) an extended concept of ToM, i.e., RF (or mentalizing) and its assessment; (ii) the influence of diverse factors on ToM performances in people with schizophrenia, such as the causal factors of ToM deficits, personal emotional trauma, state of psychosis, and psychosocial interventions; and (iii) potential reliable and valid ToM measures for evaluating ToM performances in an individual’s real social context, as compared with ToM tasks, which measure clearly defined ToM concepts or ToM dimensions in experimental contexts.” (Page 24, Lines 545–553)In Conclusions: “Our review provides clinical and research implications in several forms: guidance for practitioners and researchers in choosing from the extant ToM tasks for people with schizophrenia; a recommendation for researchers to improve ToM tasks; and suggestions for future directions of ToM studies in people with schizophrenia.” (Page 25, Lines 575–580) Thank you for noting this mistake. We have removed the paragraph Lines 554–557 with the tracked changes in the manuscript. 5.      Further readings that could be helpful and should be included in the introduction or discussion are:Dziobek, I., Fleck, S., Kalbe, E., Rogers, K., Hassenstab, J., Brand, M.,Kessler, J., Woike, J. K., Wolf, O. T., & Convit, A. (2006). IntroducingMASC: A movie for the assessment of social cognition. Journalof Autism and Developmental Disorders, 36(5), 623–636. https://doi.org/10.1007/s10803-006-0107-0 Dimaggio, G., Popolo, R., Salvatore, G., & Lysaker, P. H. (2013).Mentalizing in schizophrenia is more than just solving theory ofmind tasks. Frontiers in Psychology, 4, 83. https://doi.org/10.3389/fpsyg.2013.00083 Fonagy, P., Luyten, P., Moulton-Perkins, A., Lee, Y.-W., Warren, F., Howard,S., Ghinai, R., Fearon, P., & Lowyck, B. (2016). Development and validation of a self-report measure of mentalizing: The reflective functioning questionnaire. PLoS One, 11(7), Article e0158678. https://doi.org/10.1371/journal.pone.0158678 Richter, F., Steinmair, D., Löffler-Stastka, H. (2021). Construct Validity of the Mentalization Scale (MentS) within a mixed psychiatric sample. Frontiers in Psychology, 12:608214. doi: 10.3389/fpsyg.2021.608214. Steinmair, D., Horn, R., Richter, F., Wong, G., Löffler-Stastka, H. (2021). Mind reading improvements by mentalization based therapy training. The Bulletin of the Menninger Clinic, 85(1), 59-83. 6.      Further please add for further outlook the necessary discussion to distinguish or merge ToM and Mentalization/Reflecting functioning and add also add clinical application studies in your argumentation.Response:Thank you for noting this issue. We have read the articles which you recommended (Dimaggio et al., 2013; Fonagy et al., 2016; Richter et al., 2021) and added an explanation of this issue in the Discussion, as follows. “In this review, we noticed that the “ToM” concept had divergent operational definitions in the various studies and was assessed in different ways. Apart from the ToM ability being defined as “the ability to perceive others’ thoughts or feelings” in this study, the ToM ability can also be extended to the concept of “reflective functioning”. Reflective functioning (RF) (or mentalizing) refers to the ability to interpret the internal mental states, such as feelings or thoughts, of both the self and others. That is, RF involves both self- and other-mentalizing. The other-mentalizing ability has been included in the scope of our systematic review and defined as the ToM ability. In contrast, self-mentalizing involves the ability to infer self-mental states, which is introspective. It is inferring according to one’s internal information, such as autobiographical memory and emotions. Evidence has shown that other- and self-mentalizing are dissociable, yet interactive, in a shared mentalizing neuro-network [25]. Other-mentalizing (i.e., ToM) and self-mentalizing are different abilities and cause different responses in their respective posterior regions, including the temporoparietal junction, precuneus/posterior cingulate complex, and superior temporal sulcus in the mentalizing neuro-network. Hence, other- and self-mentalizing have been found to be responsive to different types of ToM measures. Impairment in other-mentalizing can be detected by the ToM tasks with observable reality, which is within the scope of this systematic review. In contrast, the self-mentalizing employed in RF is more responsive to the narrative-based ToM assessments [80, 81]. However, sharing the same mentalizing network, other- and self-mentalizing are interactive, providing reciprocal references for each other and contributing to RF. An individual with high RF can demonstrate some certainty about his or her own mental states and those of others, meanwhile knowing that this certainty can be modified by self-internal stored information and external information from others [82]. Given that other-mentalizing and self-mentalizing may influence each other, exploring self-mentalizing or RF in people with schizophrenia may provide a broader scope to understand their ToM difficulties.” (Pages 23–24, Lines 504–529)References:

  1. Fertuck, E.A.; Mergenthaler, E.; Target, M.;  Levy, K.N.; Clarkin, J.F. Development and criterion validity of a computerized text analysis measure of reflective functioning. Psychother. Res. 2012, 22, 298-305.
  2. Richter, F.; Steinmair, D.; Löffler-Stastka, H. Construct validity of the mentalization scale (MentS) within a mixed psychiatric sample. Front. Psychol. 2021, 12.
  3. Fonagy, P.; Luyten, P.; Moulton-Perkins, A.; Lee, Y.-W.; Warren, F.; Howard, S.; Ghinai, R.; Fearon, P.; Lowyck, B. Development and validation of a self-report measure of mentalizing: The reflective functioning questionnaire. PLoS One. 2016, 11, e0158678.
  4. Dimaggio, G.; Popolo, R.; Salvatore, G.; Lysaker, P. Mentalizing in schizophrenia is more than just solving theory of mind tasks. Front. Psychol. 2013, 4, 83.

 Thank you for the suggestion. We have added clinical application studies in the Introduction, as follow.“Fortunately, it has been evidenced that the ToM impairment in people with schizophrenia can be improved by mentalizing-based interventions [19-21]. Measuring ToM in people with schizophrenia is crucial because ToM has been identified as an important target and outcome measure for psychosocial treatment.” (Page 2, Lines 55-58)References:

  1. Vass, E.; Fekete, Z.; Simon, V.; Simon, L. Interventions for the treatment of theory of mind deficits in schizophrenia: Systematic literature review. Psychiatry Res. 2018, 267, 37-47.
  2. Vass, E.; Simon, V.; Fekete, Z.; Lencse, L.; Ecseri, M.; Kis, B.; Simon, L. A novel virtual reality-based theory of mind intervention for outpatients with schizophrenia: A proof-of-concept pilot study. Clin. Psychol. Psychother. 2020, 1-12.
  3. Steinmair, D.; Horn, R.; Richter, F.; Wong, G.; Löffler Stastka, H. Mind reading improvements in mentalization-based therapy training. Bull. Menninger Clin. 2021, 85, 59-82.

Reviewer 3 Report

Thank you for the opportunity to review this interesting paper. The rationale for this synthesis is clear.

I have a few minor points for the authors to consider.

  1. I suggest that either in the introduction, or in the discussion,  the authors acknowledge that ToM is not an unproblematic concept.  I think this complexity strengthens your argument, underscoring the importance of a clearly defined concept with strong psychometrics. See for example McCabe et al (2004) which problematises the assumption of impaired ToM in people with schizophrenia (McCabe, R., Leudar, I., & Antaki, C. (2004). Do people with schizophrenia display theory of mind deficits in clinical interactions?. Psychological Medicine34(3), 401)
  2. Line 72, interpreting irony is usually considered a pragmatic skill, in both Gricean and Neo-Gricean traditions. I suggest that metalinguistic is changed to 'pragmatic'. 
  3. Given that the authors included studies in both English and Mandarin (or both were eligible for inclusion), it would be useful to know which assessments have been translated / adapted into Mandarin.
  4. Table 2 is particularly hard to read. This may be a typesetting issue as the rows do not always align the the headings do not always fit within the cell. This may be something to explore with the journal - perhaps a landscape page is possible? This table is really the practical take away for practitioners in my opinion and so having it as readable as possible is a priority.
  5. Phrasing: I would suggest that throughout, the phrasing of "patients with SCZ" be replaced by "people with schizophrenia". 

Author Response

Response to Reviewer’s comments

Reviewer 3

Thank you for the opportunity to review this interesting paper. The rationale for this synthesis is clear.

I have a few minor points for the authors to consider.

  1. I suggest that either in the introduction, or in the discussion, the authors acknowledge that ToM is not an unproblematic concept. I think this complexity strengthens your argument, underscoring the importance of a clearly defined concept with strong psychometrics. See for example McCabe et al (2004) which problematises the assumption of impaired ToM in people with schizophrenia (McCabe, R., Leudar, I., & Antaki, C. (2004). Do people with schizophrenia display theory of mind deficits in clinical interactions?. Psychological Medicine, 34(3), 401)Response:Thank you for the suggestion and the helpful article. We have read your suggested article and added this in the Discussion, as follows.“ToM performance in the everyday contexts of people in schizophrenia has been proposed to be more complicated than the ToM ability measured by standardized ToM tasks [83]. Therefore, when interpreting the ToM impairment of people with schizophrenia in ToM tasks, several issues should be considered: (i) the causal factors of ToM deficits, e.g., the loss of neurocognitive ability and poverty of early attachment; (ii) the impact of affective states; e.g., the intense emotions of being abused may create a bias for inferring others’ mental states; and (iii) the phase of illness in which these deficits occur and the manner in which these deficits present during the psychosocial treatment. Indeed, McCabe et al. [84] proposed a different method to assess the ToM performance of people with chronic schizophrenia. Their ToM performance was evaluated by analyzing the conversational interactions between people with schizophrenia and mental health professionals in outpatient consultations and cognitive behavioral therapy. Their results showed that the interviewee might still have persecutory delusion and at the time could correctly infer the interviewer’s intentions. That is, mentalizing in everyday interactions may be triggered and framed by the behaviors of others, unlike in controlled experimental ToM tasks.” (Page 24, Lines 530–544)References:
  2. Dimaggio, G.; Popolo, R.; Salvatore, G.; Lysaker, P. Mentalizing in schizophrenia is more than just solving theory of mind tasks. Front. Psychol. 2013, 4, 83.
  3. McCabe, R.; Leudar, I.; Antaki, C. Do people with schizophrenia display theory of mind deficits in clinical interactions? Psychol. Med. 2004, 34, 401-412.
  4. Line 72, interpreting irony is usually considered a pragmatic skill, in both Gricean and Neo-Gricean traditions. I suggest that metalinguistic is changed to 'pragmatic'.Response:Thank you for noticing this issue. We have followed your suggestion and changed the word ‘metalinguistic’ to ‘pragmatic’ in the manuscript. (Page 2, Line 83)       Given that the authors included studies in both English and Mandarin (or both were eligible for inclusion), it would be useful to know which assessments have been translated / adapted into Mandarin.Response:Thank you for the inquiry. We have attempted to search for and include the Mandarin or English articles that studied the ToM tasks in schizophrenia. Of the 13 types of ToM tasks, only the type of ST (Strange Stories) has been adapted into Mandarin [58]. This Mandarin version of the Strange Stories comprises the concepts of“figure of speech”, “lies”, “white lies”, and “joke” (Table 2) . It has been evaluated for its psychometric properties, including the test–retest reliability, the concurrent validity, and the ecological validity. The results showed that all the examined psychometric properties were poor (Table 3). The reference is below. Reference:
  5. Chen, K.W.; Lee, S.C.; Chiang, H.Y.; Syu, Y.C.; Yu, X.X.; Hsieh, C.L. Psychometric properties of three measures assessing advanced theory of mind: Evidence from people with schizophrenia. Psychiatry Res. 2017, 257, 490-496.
  6. Table 2 is particularly hard to read. This may be a typesetting issue as the rows do not always align the the headings do not always fit within the cell. This may be something to explore with the journal - perhaps a landscape page is possible? This table is really the practical take away for practitioners in my opinion and so having it as readable as possible is a priority.Response:Thank you for the suggestion. We have re-formatted Table 2, as follows. (Pages 9–12) 

Table 2. Types, Concept, Construct and Administration of the Current ToM Tasks.

Name of Each Type of ToM Task (Reference)

ToM Concept

Construct

N of Included Studies

Task Content

Presentation Modality

Answer Mode

Inclusion of Control Questions/

Items

Scoring

Hinting task

V1 [47]

Infer real intentions behind indirect words

Cog & Aff

12

10 short stories about a social interaction between two characters. Each story ends with one character dropping a hint.

Verbal stories

Open-ended questions

No

0, 1, 2.

V2 [16]

Movie sequence as presentation modality.

Multisensory movie (verbal, visual, auditory)

0, 1, 2.

First order False Belief stories

V1 [48, 49]

FB1

Understand one has a false belief about the reality

Cog & Aff

17

Sally and Anne story & Cigarettes story

Verbal stories & visual adds

Open-ended questions

Q: M & C

ToM: 0, 1. Explanation: 0, 1, 2, 3. 

V2 [50]

-FB1 with deception

Understand psychological states guide behaviors & to deceive

Cog & Aff

First order FB and deception stories with series of cartoon drawings.

0, 1, 2.

Scored only when control question is correctly answered

Second order False Belief stories

V1 [49, 51]

FB2

Understand one has a false belief about the belief of another one

Cog

15

Ice-Cream Van story; Burglar story.

Verbal stories & visual adds

Open-ended questions

Q: M & C

0, 1, 2. 

V2 [50]

-FB2 with deception

Understand one ignores misinformation because another one is trying to deceive

Cog & Aff

Second order FB and deception stories with series of cartoon drawings

0, 1, 2.

Scored only when control question is correctly answered.

Table 2. (Continued)

Name of Each Type of ToM Task (Reference)

ToM Concept

Construct

N of Included Studies

Content of Task

Presentation Modality

Answer Mode

Inclusion of Control Questions/Items

Scoring

False Belief picture sequencing

V1 [31]

Correctly complete a set of pictures based on false belief inferences

Cog

14

Arrange 4 four-card picture sequences of false beliefs in a correct order.

4 types of stories: social-script, mechanical, false-belief & capture.

Visual picture sequences

Both versions: Sequence story pictures.

V2 includes an additional open-ended questions

I: Inferential reasoning ability 

0-6.

V2 [52]

6 picture stories of false beliefs & 23 questions with first & second order ToM and non-mental questions.

Picture sequencing: 0-6;

ToM questionnaire: 0, 1.

Character Intention task [53, 54]

Understand the intention of a person in subtle social cues

Cog & Aff

5

30 or 42 sets of comic strips. Each strip: 3 pictures in sequence & answer cards (Attribution of intention & Attribution of false belief).

Visual pictures

Multiple choice questions

I: Basic reasoning ability

0, 1.

Visual Jokes [55]

Detect visual jokes involving attribution of ignorance, false belief or deception

Cog

5

Two sets of 10 cartoon jokes. Set 1: Mental state attribution to false belief and deception. Set 2: Physical/behavior scene.

Visual jokes

Open-ended questions

I: Other general cognitive deficits

0, 1.

Irony task [34, 56]

Understand the opposition between literal & true meanings of words

Cog & Aff

5

9 or more stories with ironical utterance.

Verbal stories & written copy

Multiple choice questions

Q: C   

0, 1.

Faux Pas [30, 57]

Infer different perspectives: speaker’s thinking & listener’s feeling

Cog & Aff

11

FP stories with questions of recognition & understanding of FP.

Verbal stories with a print copy

Open-ended questions

I: Basic reasoning ability, attention, Q: M or C. 

0, 1.

Table 2. (Continued)

Name of Each Type of ToM Task (Reference)

ToM Concept

Construct

N of Included Studies

Content of Task

Presentation Modality

Answer Mode

Inclusion of Control Questions/Items

Scoring

Yoni's Verbal and Eye Gaze Cues [26]

Judge mental states based on verbal and eye gaze cues.

Cog & Aff

6

Each of 87 trials: a cartoon outline of a face and four colored pictures around each corner.

Questions: first-order or second-order ToM, cognitive or affective ToM.

Visual & Verbal (written questions)

Multiple choice questions

Q: Attention & C

0, 1.

Story test

V1 [35]

Multiple concepts: various, 4-5

Cog & Aff

6

All 3 versions use stories.

8 ToM: Double bluff, mistakes, persuasion, white lies.

Verbal & visual adds

Open-ended questions. 

I: Cause-effect inference

0, 1.

V2 [58]

5 ToM: Figure of speech, lies, white lies, joke.

Q: M & C

0, 1.

V3 [59]   

18 ToM stories. False beliefs, false attributions, lies, sarcasm, faux pas. 

Q: M & C

0, 1, 2.

Movie for Social Cognition

V1 [60]

Multiple concepts: 5.

Cog & Aff ToM;

Emotion perception

3

Movie for the Assessment of Social Cognition (MASC): 15 min movie about characters getting together for a dinner party: paused 46 times for 48 questions. ToM: first- and second-order false belief, faux pas, metaphor, or sarcasm.

Scenario: multi-modalities.

Question: verbal & written

Multiple-choice questions

Q: M & C,      I: Reasoning ability

0, 1. Outputs: Error categories, mental state modalities and non-social inferencing.

M & C: 1, 0.5, 0.

V2 [61]

Multiple concepts: false belief, deception, faux pas, humor, sarcasm, and persuasion

Cog & Aff ToM;

Emotion perception

Virtual Assessment of Mentalising Ability (VAMA): 12 video clips depicting a social drama within interactive virtual environment.

Interactive multi-modalities

Multiple-choice questions

I: Reasoning ability

Scored in two ways.

3-point scale: 0 (impaired), 0 (hyper) , 1 (reduced) & 2 (accurate).

Dichotomous scale: 1 (accurate), 0 (incorrect: any wrong response).

Table 2. (Continued)

Name of Each Type of ToM Task (Reference)

ToM Concept

Construct

N of Included Studies

Content of Task

Presentation Modality

Answer Mode

Inclusion of Control Questions/Items

Scoring

Reading the Mind in the Eyes Test [32]

Infer mental states from the pictures of persons’ eyes & apply affective terms

Emotion recognition; Vocabulary comprehension; Aff ToM

9

36 eye photos showing emotions. Choose one term from 4 choices. Include definitions of emotional terms for reference.

Visual photos & Verbal question

Multiple-choice questions

I: Face-recognition problems

0, 1

The "Moving Shapes" paradigm

V1 [62]

Infer intentions of silent cartoon figures enacting social drama

ToM, Social perception; Visual-spatial Problem solving

6

12 animations with two characters, a big red triangle and a small blue triangle, moving on framed white background.

Non-verbal animations

Open-ended questions

I: Alexithymia problem and empathetic ability 

4 dimensions: Intentionality: 0-5;

Appropriateness: 0-3.

Certainty: 0-3;

Length: 0-4.

V2 [63, 64]

a large triangle, small triangle and small circle enact social drama.

Both versions contain following questions.

Multiple choice questions

No

0, 1.

Note: N = numbers; Cog = Cognitive; Aff = Affective; V1 = Version 1; V2 = Version 2; Q = Question; I = Items; M = Memory; C = Comprehension; First order False belief = FB1; Second order False belief = FB2.

 5.      Phrasing: I would suggest that throughout, the phrasing of "patients with SCZ" be replaced by "people with schizophrenia".Response:Thank you for the reminder. We have re-phrased “patients with SCZ" as “people with schizophrenia” throughout this manuscript.  
